# Hypnosis-Assisted Awake Craniotomy for Eloquent Brain Tumors: Advantages and Pitfalls

**DOI:** 10.3390/cancers16091784

**Published:** 2024-05-05

**Authors:** Giulia Cossu, Alberto Vandenbulcke, Sonia Zaccarini, John G. Gaudet, Andreas F. Hottinger, Nina Rimorini, Arnaud Potie, Valerie Beaud, Ursula Guerra-Lopez, Roy T. Daniel, Chantal Berna, Mahmoud Messerer

**Affiliations:** 1Department of Neurosurgery, University Hospital of Lausanne and University of Lausanne, 1011 Lausanne, Switzerland; alberto.vandenbulcke@chuv.ch (A.V.); roy.daniel@chuv.ch (R.T.D.); mahmoud.messerer@chuv.ch (M.M.); 2L. Lundin and Family Brain Tumor Research Center, Departments of Oncology and Clinical Neurosciences, University Hospital of Lausanne, 1011 Lausanne, Switzerland; andreas.hottinger@chuv.ch; 3Department of Anesthesiology, University Hospital of Lausanne, 1011 Lausanne, Switzerland; s.zaccarini@swisspain.ch (S.Z.); john.gaudet@chuv.ch (J.G.G.);; 4Center for Integrative and Complementary Medicine, Department of Anesthesiology, University Hospital of Lausanne, The Sense and University of Lausanne, 1011 Lausanne, Switzerland; nina.rimorini@chuv.ch (N.R.); chantal.berna-renella@chuv.ch (C.B.); 5Division of Neuro-oncology, Department of Oncology, University Hospital of Lausanne and University of Lausanne, 1011 Lausanne, Switzerland; 6Service of Neuropsychology and Neurorehabilitation, University Hospital of Lausanne, 1011 Lausanne, Switzerland; valerie.beaud@psychologie.ch (V.B.); ursula.guerra-lopez@chuv.ch (U.G.-L.)

**Keywords:** anesthesia, awake craniotomy, brain neoplasms, brain tumor, eloquent area, monitored anesthesia care, hypnosis, hypno-sedation

## Abstract

**Simple Summary:**

Awake craniotomy (AC) is a neurosurgical technique that allows safe and efficient resection of brain tumors in eloquent areas. It can be completed using two anesthetic strategies: monitored anesthesia care (MAC), which relies on pharmacological sedation and analgesia, or hypnosis-assisted awake craniotomy (HAAC), during which hypnotic suggestions could reduce the need for hypnotics and opioids. In this study, we retrospectively compared the characteristics and outcomes of patients undergoing AC with either one of those anesthetic techniques. A total of 22 patients were analyzed: 14 underwent HAAC and 8 underwent AC under MAC. Compared to patients in the MAC group, those in the HAAC group received a significantly smaller quantity of hypnotics and analgesics. Although patients in the HAAC group experienced more pain, they reported lower stress levels and higher satisfaction scores. These results suggest that HAAC safely reduces the need for pharmacological agents during surgery while providing significant psychological benefits. However, further research is needed to confirm and expand this preliminary work.

**Abstract:**

Background: Awake craniotomy (AC) is recommended for the resection of tumors in eloquent areas. It is traditionally performed under monitored anesthesia care (MAC), which relies on hypnotics and opioids. Hypnosis-assisted AC (HAAC) is an emerging technique that aims to provide psychological support while reducing the need for pharmacological sedation and analgesia. We aimed to compare the characteristics and outcomes of patients who underwent AC under HAAC or MAC. Methods: We retrospectively analyzed the clinical, anesthetic, surgical, and neuropsychological data of patients who underwent awake surgical resection of eloquent brain tumors under HAAC or MAC. We used Mann–Whitney U tests, Wilcoxon signed-rank tests, and repeated-measures analyses of variance to identify statistically significant differences at the 0.05 level. Results: A total of 22 patients were analyzed, 14 in the HAAC group and 8 in the MAC group. Demographic, radiological, and surgical characteristics as well as postoperative outcomes were similar. Patients in the HAAC group received less remifentanil (*p* = 0.047) and propofol (*p* = 0.002), but more dexmedetomidine (*p* = 0.025). None of them received ketamine as a rescue analgesic. Although patients in the HAAC group experienced higher levels of perioperative pain (*p* < 0.05), they reported decreasing stress levels (*p* = 0.04) and greater levels of satisfaction (p = 0.02). Conclusion: HAAC is a safe alternative to MAC as it reduces perioperative stress and increases overall satisfaction. Further research is necessary to assess whether hypnosis is clinically beneficial.

## 1. Introduction

The primary goal of brain surgery for intra-axial tumors is to achieve maximal resection while preserving neurological functions, in order to improve survival and quality of life [1,2,3,4]. Achieving such results when tumors are located in eloquent areas is challenging. During surgery, ultrasonography, magnetic resonance imaging (MRI), neuronavigation, or fluorescence can help identifying tumor boundaries [5,6,7,8,9,10]. However, intraoperative imaging techniques do not identify functional neurological areas that should be preserved. Preoperative tractography and functional MRI are valuable tools in this context, yet they have their limitations [11]. Therefore, intraoperative brain stimulation, performed in an asleep or awake setting, remains the most reliable technique for safe resections within functional boundaries [12,13,14,15,16]. While motor mapping can be performed under general anesthesia [15,17], awake testing is required to map brain areas involved in language, emotions, and cognitive functions [12].

Thus, awake craniotomy (AC) is recommended for resection of tumors in eloquent or near-eloquent areas, as it is associated with maximal extent of resection and improved neurological outcomes [18,19,20,21]. From an anesthetic standpoint, AC is traditionally completed using either the asleep-awake-asleep technique (AAA) or the monitored anesthesia (MAC) technique. While AAA offers better airway control, is it associated with a greater risk of emergence delays which may affect the quality of neuropsychological testing [22]. Both techniques rely on the administration of hypnotic and opioid agents. They have both been linked to intraoperative discomfort, anxiety, and postoperative stress or neuropsychological sequelae [23,24,25,26,27,28,29]. Hypnosis-assisted AC (HAAC) is an emerging alternative that may not only reduce the need for pharmacological sedation and opioid analgesia [30,31], but also improve the psychological recovery of patients after surgery [31,32,33,34].

We have experience of using both MAC and HAAC techniques in our institution since 2021. The aim of this study is to compare the management and outcomes of patients who underwent awake resection of brain tumors located in or near eloquent areas using MAC or HAAC. 

## 2. Materials and Methods

We retrospectively analyzed the demographic, medical, radiological, anesthetic, surgical, and postoperative data of all adult patients who were admitted for resection of an eloquent or near-eloquent brain tumor under AC at our institution between June 2021 and May 2023. We excluded patients with a Karnofsky performance status (KPS) score < 80 and presenting comorbidities that precluded the safe completion of AC, along with those who denied consent for research. The study was approved by our local research ethics board (CER-VD 2023-01516).

### 2.1. Preoperative Anesthetic Management

At least 2 weeks before surgery, the anesthesiologist provided thorough information about the awake craniotomy procedure irrespective of whether it was going to be performed under hypnosis or MAC. In addition to collecting the medical history, the discussion aimed to characterize patients’ motivations, observational capacity, and ability to understand and memorize tasks, but also their favorite activities along with the type of music and situations they may find relaxing. The anesthetic technique was selected based on clinical criteria, patient preference, and availability of trained personnel. After completing a practice hypnosis session using safe place imagery, candidates for hypnosis were invited to ask any questions they may have. Upon admission, all patients received further photo-illustrated explanations about the course of surgery. Patients did not receive any anxiolytic premedication.

### 2.2. Intraoperative Anesthetic and Surgical Procedures

The anesthesiologist performed a scalp block by infiltrating 30 to 40 mL of ropivacaine 0.5% with epinephrine 1:400,000 at the supratrochlear, supraorbital, zygomaticotemporal, and auriculotemporal nerves, as well as the greater and lesser occipital nerves. Hypnotics and analgesics were administered using a dedicated intravenous catheter. A radial arterial catheter was inserted to monitor blood pressure continuously during surgery. Urinary catheters were avoided. Scalp and peripheral leads were placed by a clinical electrophysiologist. 

Following head placement in a Mayfield clamp with infiltration of fixation points, patients were encouraged to participate to reach a comfortable supine position on the operating table with their head turned to face the anesthesiologist and the neuropsychologist. The incision site was infiltrated with local anesthesia prior to surgical start.

We used a monopolar probe to map cortical and subcortical functional areas using 500-msec monophasic stimuli with 3-msec interstimulus intervals at intensities ranging between 1.5 and 3.5 mA. We also used a stimulating aspiration to assess functional boundaries in the subcortical area [35]. Recurring positive mapping signals in the cortical or subcortical areas identified limits of resection. Neuropsychological assessment was performed continuously during resection using a panel of tasks defined preoperatively. The extent of resection was adjusted according to the tests’ results, with the aim of achieving supramarginal resection. Once neuropsychological testing was no longer necessary, sedation or hypnosis resumed until the end of the surgical procedure. 

### 2.3. Specificities of HAAC Technique

A hypnosis-trained physician initiated the modulation of perceptions, sensations, and emotions by using permissive suggestions one hour before surgery. Prior to patient transfer to the operating room, lights were dimmed and staff were requested to use a soft voice to communicate. Access to the operating room and interaction with the patient were restricted to optimize the hypnotic state. 

Invasive procedures such as catheter insertions or head frame placement were completed using distraction or suggestion techniques. The hypnotic state was then enhanced with multi-sensory suggestions shortly before surgical start. Once the dural layer was opened, the hypnotic state was reversed to facilitate neuropsychologist testing.

We used a low-dose remifentanil infusion (0.01–0.05 mcg/kg/min) with continuous capnographic monitoring to optimize analgesia during surgery. If necessary, we combined it with low-dose dexmedetomidine (0.3–0.8 mcg/kg/hr) to reinforce analgesia while providing mild sedation [36]. We used propofol to achieve deeper sedation after the neuropsychological testing in selected cases only. 

### 2.4. Specificities of MAC Technique

For sedation, we used varying rates of intravenous propofol (20–100 mcg/kg/min) throughout the procedure. Propofol was stopped on incision of the dural layer. At the end of the neuropsychological testing phase, propofol sedation was resumed and combined with a dexmedetomidine infusion (0.5–1.0 mcg/kg/hr) when necessary. 

For analgesia, we combined a low-dose remifentanil infusion (0.01–0.05 mcg/kg/min) with a scalp block as previously described. Low-dose ketamine (10 mg administered slowly over a 10-minute window) was administered as rescue, if necessary, after tumor resection only. 

### 2.5. Postoperative Management, Neuropsychological Tests, and Periprocedural Questionnaires

All patients had an early postoperative 1.5 or 3T MRI (Siemens, Erlangen, Germany) control within 48 hours after surgery to evaluate the extent of resection, which was defined as gross total resection (GTR) when no residual tumor could be identified on 3D T1-weigthed sequences with gadolinium administration and T2-weighted sequences for non-contrast enhancing tumors. 

A trained neuropsychologist was in charge of completing baseline and follow up neuropsychological assessments one day before surgery and 2 to 3 days after surgery, respectively. 

We administered the following questionnaires at specific timepoints to evaluate the perception and psychological impact of surgery: -the Hospital Anxiety and Depression Scale (HADS) [37] and the Perceived Stress Scale (PSS) [38] were used to assess depression, anxiety, and perception of stress before surgery;-a Visual Analog Scale (VAS) was used to quantify stress, comfort, and pain preoperatively, during surgery, and at the end of the procedure;-the Impact of Events Scale-Revised (IES-R) [39], the Peritraumatic Distress Inventory (PDI) [40], and the Peritraumatic Dissociation Experiences Questionnaire (PDEQ) [41] were used 3 to 7 days after the procedure to retrospectively assess the psychological impact and level of dissociation at the time of surgery;-a series of 19 questions designed by our team to assess the level of satisfaction with surgical, anesthetic, and neuropsychological management, identify painful memories during the surgical procedure, test orientation in time and space, and establish whether patients would undergo a second surgery under similar conditions (Appendix A).

### 2.6. Statistical Analysis

We stored and assembled all data in a coded database. Once extracted, data were analyzed with Stata/IC 17 software (StataCorp, College Station, TX, USA). Continuous variables were assessed visually and tested for parametric distribution using the Kolmogorov–Smirnov test. They were summarized as means ± standard deviation (SD). We computed frequencies and proportions to summarize categorical variables.

To compare continuous variables between the hypnosis and the sedation groups, we used Mann–Whitney U tests for univariate analyses. Wilcoxon signed-rank and ANOVA tests for repeated measures were used to compare VAS trends over time in both groups. Point-biserial correlation was used to analyze whether a difference existed between continuous variables. The phi coefficient was used to measure the association between two dichotomous variables. Fisher’s exact test was used for comparison of categorical variables. The significance level was set at a *p*-value < 0.05.

## 3. Results

Twenty-two of twenty-eight eligible patients gave their consent for research: fourteen patients underwent AC under hypnosis, and eight under MAC. 

The median age was 46.5 years (IQR: 36.5–53.5 yo). The gender distribution was evenly split, with 11 women and 11 men in the cohort. Demographic characteristics were similar between the two groups (Table 1). In the HAAC group, two patients had a history of anxio-depressive disorder. The sample included nine cases of World Health Organization (WHO) grade 2 glioma, 3 WHO grade 3 and 8 WHO grade 4 according to the 2021 WHO classification [42]. Two patients presented with non-glioma tumors: one with high grade neuroepithelial cancer, one with a metastasis of a lung adenocarcinoma. Four patients had previously been operated on with an AAA or MAC protocol and three under general anesthesia at other institutions. None of the patients required a conversion to general anesthesia. 

Total time spent in the operating room was shorter by 61 min on average in the HAAC group, but this difference was not statistically significant (mean 291 min, SD +/−59 min; vs. 352 min, SD +/−75 min in the MAC group; *p* = 0.06). This was in part due to surgery being shorter by an average of 50 min in the HAAC group (mean of 196 min vs. 246 in the MAC group; *p* = 0.08) (Figure 1).

Tumor volume was lower in the HAAC group, but the difference was not statistically significant (Table 1). Complete resection was achieved in 9/14 patients in the HAAC group (64%) versus 4/8 patients in the MAC group (*p* = 0.6), as the resection was stopped with repetitive positivity of the cortical or subcortical mapping. No intraoperative seizures were reported. In the HAAC group, neuropsychological testing was interrupted in two cases (14%) due of excessive patient fatigue. Postoperatively, we detected transient speech and/or motor deficits that fully recovered on the first follow-up visit at 6 weeks after surgery. 

As expected, anesthesia management was significantly different between the two groups. Further details are provided in Table 2. 

There were significant differences in the use of analgesics (Figure 2): patients in the HAAC group received significantly less remifentanil than patients in the MAC group (436 mcg vs 659 mcg; *p* = 0.047). As average infusion rates in the two groups were similar, this difference was mainly due to shorter administration durations in the HAAC group (not significant; *p* = 0.059). Patients in the HAAC group also received less rescue analgesic therapy (7/14 patients vs 8/8 patients in the MAC group; *p* = 0.02, Figure 2) such as low-dose ketamine, which was used in the MAC group only. 

There were also significant differences in the use of hypnotic agents (Figure 3). While all patients in the MAC group were sedated using propofol, only 4/14 (28.6%) in the HAAC group received propofol (*p* = 0.002). Dexmedetomidine was used similarly in the two groups, but the average total dose was higher in the HAAC group (0.71 mcg/kg/h vs. 0.31 mcg/kg/h for the MAC group; *p* = 0.025), in part because more patients in the HAAC group received an activation bolus. Such differences certainly contributed to the use of norepinephrine for vasopressor support in the MAC group only (*p* = 0.0017). 

Table 3 summarizes the clinical data collected in the perioperative period.

There was no statistically significant difference between the two groups in terms of depression and anxiety evaluated with HADS and PSS questionnaires before surgery. 

Average VAS values for stress were similar between the two groups. However, trends over time were significantly different between the two groups. More specifically, patients in the HAAC group reported decreasing stress scores over time, reaching the lowest values postoperatively (*p* = 0.04), whereas patients in the MAC group reported increasing stress scores over time, reaching the highest values postoperatively (*p* = 0.09; Figure 4). Gender and age did not appear to affect perioperative stress levels. 

Pain scores were low in general, with patients in the HAAC group reporting higher average scores than those in the MAC group (mean perioperative pain scores: 3.8 in the HAAC group vs. 0.9 in the MAC group; *p* = 0.03). Maximal perioperative pain scores were also significantly different: 7.2 in the HAAC group vs. 2.7 in the MAC group (*p* = 0.009). When asked to identify painful moments, a majority of patients mentioned completion of the scalp block at the beginning of surgery (*n* = 4) and skin closure at the end surgery (*n* = 4). Other patients mentioned placement of the Mayfield clamp (*n* = 3) and skull opening (*n* = 3). Nevertheless, when asked postoperatively about general perioperative discomfort (VAS), there was no difference between the two cohorts. 

IES-R, PDI, and PDEQ scores were also similar in both groups, suggesting that anesthetic techniques did not affect the incidence of traumatic stress disorders. IES-R scores were significantly higher in patients with a history of anxio-depressive disorders (*p* = 0.002). 

Overall, patients in the HAAC group were more satisfied with the anesthetic management than those in the MAC group (*p* = 0.04). However, the only three patients who said they would refuse to undergo awake surgery with the same anesthetic technique belonged to the HAAC group (*p* = 0.2). 

## 4. Discussion

Awake brain mapping is associated with decreased neurological deficits and an increased extent of resection in eloquent gliomas [13,43]. This technique uses reversible functional impairments during perioperative testing to identify areas that should be protected and requires the active collaboration of the patient at crucial surgical timepoints to limit permanent functional sequelae. We have successfully completed awake craniotomies under both hypnosis (HAAC: hypnosis-assisted awake craniotomy) and pharmacological sedation (MAC: monitored anesthesia care) since 2021 in our institution. This report summarizes our observations using both techniques over a period of two years. Our findings suggest that hypnosis might reduce the total dose of hypnotics and opioids, thereby minimizing their side effects without compromising the overall comfort of patients, while possibly shortening the surgical procedures. 

In a recent meta-analysis, MAC was associated with lower AC failure rates and shorter procedure times compared with the asleep-awake-asleep (AAA) anesthetic technique [44]. Nevertheless, during MAC, anesthesiologists face challenges in ensuring patient safety, comfort, and anxiety and pain control, while maintaining wakefulness during cortical and subcortical mapping. An adequate balance should be found when performing MAC procedures, as oversedation can lead to noncooperation, hypoxia, hypercarbia, and respiratory depression, while undersedation can lead to anxiety and pain [45]. As a result, negative symptoms such as intraoperative pain, discomfort, and anxiety, as well as post-traumatic stress disorders, are described in about one third of cases [23,24,26,27,46,47]. Hypnosis was introduced as an adjuvant technique to reduce the incidence and severity of these negative symptoms [31]. Here, we confirm that HAAC is a feasible and safe technique and it should be considered as a viable alternative to standard techniques such as MAC. Although the anesthetic technique did not have a significant impact on surgical outcomes in our study, the HAAC technique was associated with shorter procedures. Our results contrast with those reported by Pesce et al., who observed longer surgical durations and longer neuropsychological testing phases in patients who received intraoperative hypnosis [48]. In our study, shorter procedure times may be partly explained by trend-level smaller tumoral volumes in the HAAC group. Additionally, we observed shorter delays until the patient was ready for intraoperative testing in the HAAC group, reflecting the lesser need for anesthetic agents under hypnosis. Indeed, patients in the HAAC group received less remifentanil and less propofol, two pharmacological agents that can cause emergence delays and respiratory depression with hypoxia, hypercarbia, and resulting brain swelling. The ability of hypnosis to ensure adequate comfort while reducing exposure to hypnotics and opioids was previously demonstrated by Hansen et al. in their description of a cohort of 50 cases of hypnosis-assisted awake craniotomies [34]. Interestingly, the average dose of dexmedetomidine was higher in the HAAC group. This drug exerts its hypnotic action through activation of central pre- and postsynaptic a2-receptors in the locus coeruleus, inducting a state of unconsciousness similar to natural sleep, with patients remaining easily rousable and cooperative [49]. However, in our cohort, dexmedetomidine was not administered systematically, as only 50% of patients received it, and did not appear to affect reported stress, comfort, and anxiety levels. 

Hypnosis may also help reduce perioperative stress, as shown by the lower acute postoperative stress scores in the HAAC group. Our data did not show differences on measures of post-traumatic stress disorder between the two groups and such results are in line with previous reports suggesting intraoperative hypnosis [31]. Interestingly, patients under hypnosis reported more pain but similar overall comfort levels during surgery to patients in the MAC group. This suggests hypnosis efficiently modulates the sensory perception and interpretation of experiences, as previously demonstrated in other surgical contexts [50].

Both sedation technique and hypnosis require fluid and strong communication between neurosurgeons and neuro-anesthesiologists, as the different surgical steps should be integrated in the anesthesiology technique during AC. For hypnosis, a trained anesthesiology team member, a willing environment, and a strong therapist–patient–surgeon relationship, as well as active patient participation and motivation, are all crucial factors for success [51]. This active participation in care, with a high level of connection between the patient and their caregivers, could explain the higher patient satisfaction in the hypnosis group. 

In conclusion, our data support the use of hypnosis as a time-efficient and reliable technique to maintain comfort and relieve stress during AC. 

## 5. Limitations

This study is a retrospective analysis of a small cohort of patients; it is not powered to draw definitive conclusions. Patients in the HAAC group tended to have lower grade and lower volume tumors compared to those in the MAC group, although the differences were not statistically significant. This could at least partly explain the shorter surgery duration observed in the HAAC group. Surgery being shorter certainly also contributed to the observed reduction in total opioids in the HAAC group.

Furthermore, we report punctual results recorded clinically in the perioperative period, not long-term evaluations. Larger prospective assessments including longer postoperative follow-up are required to evaluate the actual incidence of post-traumatic stress disorder and the long-term impact of awake craniotomy with either anesthetic technique. 

## 6. Conclusions

Hypnosis represents an interesting alternative to pharmacological sedation in the performance of awake craniotomies for eloquent brain tumor resections. The use of hypnotics and analgesics can be reduced while providing significant stress relief.

Further larger studies are necessary to assess the broader and longer-term impact of this technique, as the potential beneficial effects of hypnosis on awake craniotomies remain underexplored.

## Figures and Tables

**Figure 1 cancers-16-01784-f001:**
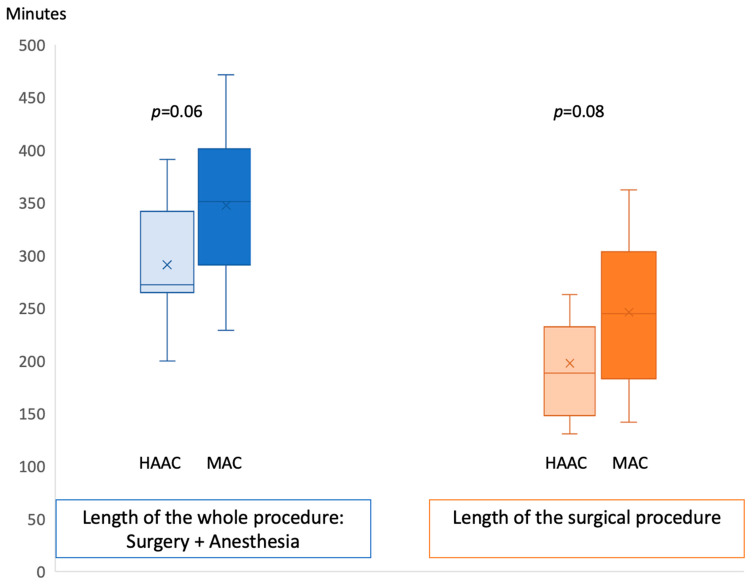
Surgical and procedural durations were both marginally shorter in the hypnosis-assisted awake craniotomy group (HAAC), compared to the monitored anesthesia care group (MAC). This trend was not statistically significant. Abbreviations: HAAC: hypnosis-assisted awake craniotomy; MAC: monitored anesthesia care.

**Figure 2 cancers-16-01784-f002:**
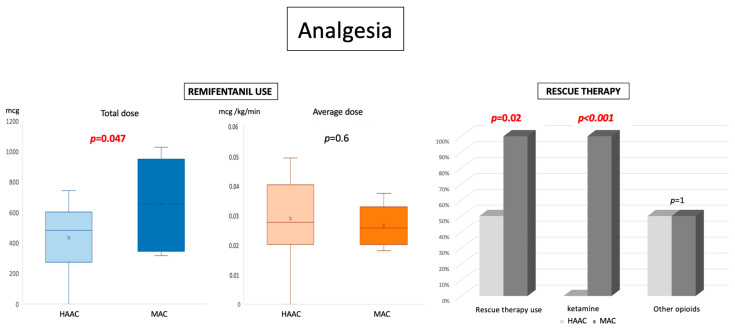
Differences in analgesia protocols between hypnosis-assisted awake craniotomies (HAAC) and monitored anesthesia care (MAC) procedures. HAAC patients received less remifentanil (436 mcg vs. 659 mcg in the MAC group; *p* = 0.047), probably mainly due to a shorter surgical duration, as the average infusion rate was not significantly different between the two groups. Rescue therapy was used less frequently in the HAAC group (*p* = 0.02). Other opioids were used similarly in the two groups. Abbreviations: HAAC: hypnosis-assisted awake craniotomy; kg: kilogram; MAC: monitored anesthesia care; mcg: microgram.

**Figure 3 cancers-16-01784-f003:**
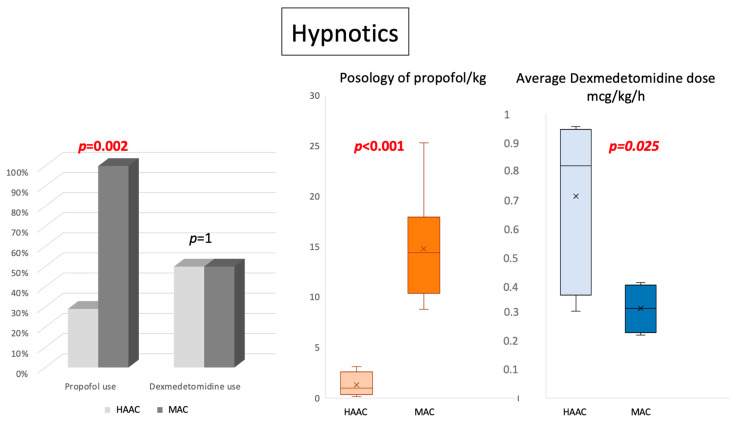
Propofol was used less often (*p* = 0.002), and at significantly lower infusion rates (p < 0.001), in the hypnosis-assisted awake craniotomy (HAAC) group. The total dose of dexmedetomidine was similar in the two groups, but average infusion rates were higher in the HAAC group (*p* = 0.025). Abbreviations: HAAC: hypnosis-assisted awake craniotomy; h: hour; kg: kilogram; MAC: monitored anesthesia care.

**Figure 4 cancers-16-01784-f004:**
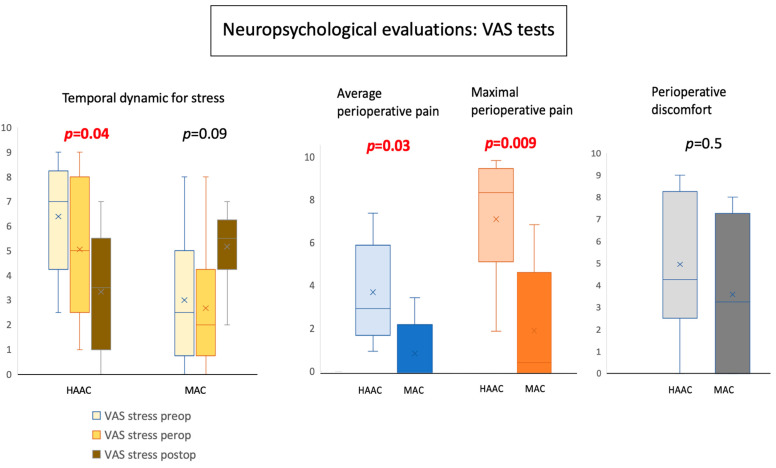
Perioperative stress measured using a visual analog scale (VAS). Patients in the HAAC group showed decreasing stress levels over time (*p* = 0.04), as opposed to patients in the MAC group, who showed increasing stress levels over time (*p* = 0.09). Pain scores were higher in the HAAC group: mean perioperative pain scores were 3.8 in the HAAC group vs. 0.9 in the MAC group (*p* = 0.03), while maximal perioperative pain scores were 7.2 in the HAAC group vs. 2.7 in the MAC group (*p* = 0.009). Comfort scores were similar in both groups. Abbreviations: HAAC: hypnosis-assisted awake craniotomy; MAC: monitored anesthesia care; preop: preoperative; perop: during surgery; postop: postoperative; VAS: visual analog scale.

**Table 1 cancers-16-01784-t001:** Demographic, clinical, and surgical data in both groups.

Variable	HAAC Cohort (14 Patients)	MAC Cohort (8 Patients)	*p*-Value
Mean age +/− SD	42.7 years +/− 9.3	49.9 years +/− 11.1	0.12
Women	7 (50%)	4 (50%)	1
Tumor localization: - Frontal - Fronto-insular - Temporal - Temporo-insular - Insular - Parietal Left hemisphere	8 1 1 1 2 1 12	2 1 3 2 0 0 8	0.17 1 0.11 0.53 0.51 1 0.51
Diagnosis: - Grade 2 glioma - Grade 3 glioma - Grade 4 glioma - Other diagnosis	7 2 3 2	2 1 5 0	0.38 1 0.08 0.51
Recurrent tumor - Previous awake craniotomy	4 (28.6%) 1/4	3 (37.5%) 3/3	0.67 0.14
Median KPS +/− IQR	100 +/− 10	90 +/− 10	0.6
Complete resection	9 (64.3%)	4 (50%)	0.66
Tumor volume +/− SD	38.7 cm^3^ +/− 40.2	60.6 cm^3^ +/− 68.3	0.43
Mean length of the surgical procedure (with anesthesia time) +/− SD	291 min, +/− 59	352 min, +/− 75	0.06
Surgery time +/− SD	196 min +/− 46.2	246 min +/− 74.6	0.08
Anesthesia time +/− SD	43 min +/− 32.5	57.7 min +/− 7.8	0.25
Intraoperative events: - Seizures - Excessive fatigue	0 2 (14.3%)	0 0	1 0.25

Abbreviations: IQR: interquartile range; HAAC: hypnosis-assisted awake craniotomy; MAC: monitored anesthesia care; min: minutes; SD: standard deviation.

**Table 2 cancers-16-01784-t002:** Detailed anesthetic management in both groups.

Variable	HAAC Cohort	MAC Cohort	*p*-Value
**Analgesia** - **Remifentanil** ○ **Mean total dose +/−SD** ○ **average dose/kg/min +/−SD** ○ **mean duration +/−SD** - **Rescue therapy:** ○ **with ketamine** ○ **with other opioids**	436 mcg, +/− 205 0.029 mcg/kg/min +/− 0.013 210 min, +/− 88 7/14 0/14 7/14	659 mcg, +/− 289 0.026 mcg/kg/min +/− 0.006 288min, +/− 89 8/8 8/8 4/8	**0.047** 0.63 0.059 **0.02** **<0.00001** 1
**Hypnotics** - **Propofol** ○ **Mean total dose +/−SD** ○ **Average dose +/−SD** - **Dexmedetomidine** ○ **Mean total dose +/−SD** ○ **Average dose / hour +/−SD**	4/14 96.2 mcg, +/− 85.5 1.3 mcg/kg; +/− 1.2 7/14 93 mcg, +/− 53 0.71 mcg/kg/h, +/− 0.3	8/8 1238 mcg, +/− 346 14.8 mcg/kg, +/− 5.3 4/8 128 mcg, +/− 54 0.31 mcg/kg/h, +/− 0.1	**0.002** **0.00008** **<0.0001** 1 0.32 **0.025**
**Noradrenaline**	0/14	4/8	**0.0017**

Abbreviations: HAAC: hypnosis-assisted awake craniotomy; MAC: monitored anesthesia care; SD: standard deviation.

**Table 3 cancers-16-01784-t003:** Results of the different neuropsychological tests and questionnaires.

Variable	HAAC Cohort	MAC Cohort	*p*-Value
**Preoperative depression confirmed at HADS** **Preoperative anxiety confirmed at HADS**	2/11 (18%) 7/11 (63.6%)	0/7 (0%) 2/7 (28.6%)	0.52 0.33
High perceived stress preoperatively (PSS) Mean VAS for preoperative stress +/− SD Mean VAS for perioperative stress +/− SD Mean VAS for postoperative stress +/− SD	0/10 (0%) 5.8 +/− 2.8 5.4 +/− 3.01 3.54 +/− 2.44	1/6 (17%) 3 +/− 2.83 2.7 +/− 2.8 5.2 +/− 1.72	0.37 0.09 0.09 0.15
Mean perioperative pain (mean +/− SD) Maximal perioperative pain (mean +/− SD)	3.8 +/− 2.32 7.25 +/− 2.8	0.9 +/− 1.52 2.7 +/− 2.86	**0.03** **0.009**
Mean perioperative discomfort (mean +/− SD)	4.9 +/− 3.2	3.6 +/− 3.8	0.5
**Posttraumatic events distress:** Pathological IES-R Pathological PDI Pathological PDEQ	3/10 (30%) 1/10 (10%) 6/10 (60%)	0/7 (0%) 2/8 (25%) 4/8 (50%)	0.2 0.5 0.7

Abbreviations: HADS: Hospital Anxiety and Depression Scale; IES-R: Impact of Events Scale-Revised; PDEQ: Peritraumatic Dissociation Experiences Questionnaire; PDI: Peritraumatic Distress Inventory; PSS: Perceived Stress Scale; SD: Standard Deviation; VAS: Visual Analog Scale.

## Data Availability

Data are contained within the article.

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
