# Peer review of "Hypnosis-Assisted Awake Craniotomy for Eloquent Brain Tumors: Advantages and Pitfalls"

_cancers, 2024, doi:10.3390/cancers16091784_

Round 1

Reviewer 1 Report

Comments and Suggestions for Authors

We carefully reviewed the presentarticle entitled “Hypnosis-assisted Awake Craniotomy for Eloquent Brain Tumors: Advantages and Pitfalls” in which Authors provide their experience in the removal of tumors in eloquent areas through the assistance of hypnosis in anestesia. The article is very interesting and well written. However, we suggest to add some information about the real efficacy of hypnosis versus hypnotic agents, which is controversial in literature. In fact, the need for an augmented quantity of dexmetedomidine in the HAAC group seems  to be strongly related to a better pharmacological performance rather than hypnosis alone. This could be important in the evaluation of an improved MAC.

Reviewer 2 Report

Comments and Suggestions for Authors

In this retrospective study, the authors assessed comparatively two techniques of awake craniotomy (AC) for resection of tumors situated in eloquent areas of the brain. Specifically, the authors looked at how the emerging hypnosis assisted AC (HAAC) technique compares to the more standard monitored anesthesia care (MAC) procedure in patients with eloquent tumors. The latter procedure has been linked to intraoperative discomfort, anxiety and post-operative stress or long-term neuropsychological sequelae. Therefore, the rationale behind the study was to identify alternative approaches for AC with the aim to reduce the need for sedatives and painkillers in this patient population while providing a good level of perioperative comfort and longer-lasting phycological benefits. One caveat of the study is that this retrospective analysis was conducted on a relatively small cohort of 22 patients (with 14 patients undergoing HAAC and 8 MAC) with a variety of tumor grades.  No data is provided regarding the performance scores (i.e., KPS) of these patients. Nonetheless, the key findings of the study were: (1) patients in the HAAC group required less pharmacological sedation (i.e., propofol), less analgesia (i.e., remifentanil) and no rescue therapy (i.e., ketamine) compared to the MAC group, (2) while the patients in the HAAC group generally experienced more pain perioperatively, they also reported comfortable levels of stress during this time interval and a higher satisfaction overall with the experience. The authors conclude that the HAAC technique appears to be a safe option for the surgical management of eloquent brain tumors that may not only reduce the need for pharmacological sedation and opioid analgesia but also potentially provide better long-term phycological outcomes for these patients. 

While I found the methodology behind the study to be generally sound and the study overall quite informative, I have a few comments for the authors.     

1.   The evidence presented does not robustly support the finding that HAAC may lead to a reduction in opiate use in this patient population.  Based on the clinical data provided, the patients in the HAAC cohort tended to have lower grade tumors compared to the ones in the MAC cohort. This could explain the differences in tumor volume (i.e., lower volumes in the HAAC cohort) and surgery time (less time spend in surgery for the patients in the HACC cohort) between the two groups. These factors could better explain the observed reduction in the need for opiates for the HACC cohort and the recovery performance of these patients. The borderline statistical significance pertaining opiate use between the cohorts could be easily explained by these factors. 

2.   In light of the above, the conclusion of the authors from lines 306-308 “Our findings confirm that hypnosis allows to reduce the reliance on sedation and opioids, thus minimizing related complications, possibly accelerating surgical times, while maintaining patient comfort and collaboration during surgery” is not entirely in agreement with the evidence provided by the authors and should be further amended.  It might be that such conclusion is more appropriate for patients with lower grade tumors and possibly better KPS scores. Although not data is provided regarding the latter. 

3.     There are a few typos throughout the manuscript that need to be fixed.  
